# A Low-Profile, Triple-Band, and Wideband Antenna Using Dual-Band AMC

**DOI:** 10.3390/s23041920

**Published:** 2023-02-08

**Authors:** Rafael Gonçalves Licursi de Mello, Anne Claire Lepage, Xavier Begaud

**Affiliations:** LTCI, Télécom Paris, Institut Polytechnique de Paris, 91120 Palaiseau, France

**Keywords:** multiband, wideband, antenna, artificial magnetic conductor, metasurface

## Abstract

When a wideband antenna is backed by an artificial magnetic conductor (AMC) reflector, the bandwidth is reduced. With the optimization of the shape of the AMC it is possible to exhibit multiband behavior, but the problem becomes complex if the bands are also intended to be wide. In this study, a methodology that exploits both the expected in-band and out-of-band behaviors of a dual-band AMC was used to design a low-profile, triple-band, and wideband directive antenna. The methodology was validated with a prototype suitable for the European standards of 4G/5G and Wi-Fi 2.4/5/6E, operating within the following bands: 2.4–2.7 GHz, 3.4–3.8 GHz, and 5.17–6.45 GHz. The measured results showed respective peak values of 8.0, 9.1, and 10.5 dBi for the broadside realized gain, front-to-back ratios larger than 19 dB, cross-polarized levels lower than -18 dB, and stable half-power beamwidths within each band. Furthermore, 3 dB gain bandwidths of 34.4%, 19.7%, and 31.0% were also measured.

## 1. Introduction

A new standard in mobile communications, which is expected to revolutionize mankind’s way of life, is today being deployed all around the world. The fifth generation of cellular networks (5G) is considered a key technology for the enabling of a series of communication capabilities such as continuous broadband experience, personalized mass market media and gaming, remote signal monitoring and machine control, and smart urban mobility [1]. The arrival of this standard is accompanied by the beginning of operation of Wi-Fi 6E, which aims at increasing the spectral efficiency per area of high-density small cells [2]. These standards will coexist alongside older ones, such as the second, third, and fourth generations of cellular networks (2G/3G/4G) and Wi-Fi 2.4/5. Together, they will constitute a heterogeneous network, characterized by a coverage tier providing wide-area coverage and mobility support, and a tier of hotspots that aims at offering high throughput within small cells while offloading the traffic from the coverage tier [3]. This scenario represents a challenge in terms of antenna technology as obtaining stable, unidirectional, high-gain patterns in multiple, wide bands using a single antenna is not trivial.

One way to obtain a unidirectional, high-gain radiation pattern with planar antennas is to add a parallel surface made of a good electrical conductor that can be considered to be a perfect electric conductor (PEC). The PEC reflector ensures most of the radiation towards a single hemisphere and avoids most of the radiation from the other hemisphere, providing thereby a low front-to-back ratio. Such a reflector maximizes the gain in the broadside direction if it is placed at h=λ/4 (λ is the wavelength) from the radiating element, thus causing the phase difference $\varphi_{t}=\varphi_{2}-\varphi_{1}$ between the reflected and direct electric fields to equal −2π in the broadside direction, as depicted in Figure 1a. Yet, the condition $-120{}^{\circ}+2N\pi <\varphi_{t}<+120{}^{\circ}+2N\pi$, where N∈ℤ, is sufficient to enhance the broadside gain [4]. The phase difference $\varphi_{t}$ accounts for the reflection coefficient phase $\varphi_{r}$ of the surface and the path phase delay $\varphi_{p}\left(f\right)=-2kh$, related to the round trip between the radiating element and the reflector, where k=2π/λ is the wavenumber. However, $\varphi_{p}$ depends on the frequency f and imposes bandwidth limitations.

Theoretically, a perfect magnetic conductor (PMC) surface with $\varphi_{r}=0{}^{\circ}$ separated by h≪λ from a radiating element provides a frequency-independent phase difference $\varphi_{t}≅0{}^{\circ}$, as represented in Figure 1b. However, the PMC behavior occurs only in special conditions, for instance, at specific frequencies in metasurfaces called artificial magnetic conductors (AMCs) [5]. Traditionally, AMCs are composed of a frequency selective surface (FSS) over a ground plane. For lossless material, their reflection coefficient magnitude equals one but their reflection phase $\varphi_{r}$ depends on the frequency f [6]. As such, the gain enhancement is again conditioned by $-120{}^{\circ}+2N\pi <\varphi_{t}<+120{}^{\circ}+2N\pi$ and presents bandwidth limitations [4]. Some works studied different AMC unit cells to widen their bandwidth, defined as $-90{}^{\circ}<\varphi_{r}\left(f\right)<+90{}^{\circ}$. It was seen that the lower the frequency, the harder it is to obtain reasonable bandwidths. For instance, bandwidths of 55.4% and 50.1% were associated with resonant frequencies of 15.7 and 8.8 GHz in [7,8] and bandwidths of 40.9%, 40.0%, and 19.4% were related to resonant frequencies of 5.7, 6.2, and 6.0 GHz, respectively, in [9,10,11]. Multi-layered AMCs were proposed in [12,13,14] to decrease the resonant frequencies down to 0.3 GHz, while achieving modest bandwidths and losing the low-profile advantage of AMCs. One interesting capability of AMCs that was recognized early on when they began to be used is the possibility of multi-band operation [15], when the PMC behavior is obtained at multiple frequencies. However, controlling the width and the spacing between the operating frequency bands is challenging.

Another kind of metasurface that can work as a reflector is the FSS without a ground plane. These structures act as stopband filters, allowing or blocking the transmission of waves at specific frequencies at resonance. In [16], it was shown that the gain of a microstrip antenna array for sub-6 GHz 5G applications, working from 3.5 to 5.8 GHz, was enhanced by 4.4 dB when it was backed by an FSS. In [17], an exhaustive review regarding the application of FSSs at ultra-wideband frequencies was presented. In [18], the reflection phase $\varphi_{r}$ of an FSS and its spacing with a dual-band antenna are optimized to simultaneously improve the gain and the bandwidth at both frequency bands. However, since these structures do not present a ground plane and allow the transmission at some frequencies, they can be inappropriate to work plated on larger structures and can even allow high levels of interference on other devices at these frequencies.

One common approach to obtain multiband, directive antennas is to address the diverse frequency bands with different radiating elements, each of which presenting a specific distance to a common reflector. A couple of filtering antennas are used as elements in a side-by-side disposition in [19] for Digital Cellular System (DCS) and Wideband Code Division Multiple Access (WCDMA). This disposition does not contribute to compactness and does not allow the elements to be over the central axis of the reflector, leading to asymmetric patterns. In fact, a simple solution exists to avoid asymmetry problems, which consists of disposing a centered antenna for the lower band surrounded by a couple of antennas for the upper band, as seen, for instance, in [20,21] with application to 2G/3G/4G. In [22], the same standards are addressed but now with a Fresnel lens covering two broadband dipoles over a U-shaped reflector. This configuration achieves a good pattern stability in both bands, but at the cost of a large profile. In [23], a quad-band magneto-electric dipole, consisting of four Γ-shaped structures, provides a moderate gain ≤5.5 dBi for 2G/3G/4G/5G, WLAN, and WiMAX. A magneto-electric antenna is also used in [24], but now associated with a metasurface with an I-shaped unit cell for 5G/WiMAX/WLAN/ X-band. A good gain is achieved but the design is complex. First, the electric and magnetic antennas must be correctly balanced to result in a unidirectional pattern. Next, a fork-shaped feeding must be adjusted to properly couple energy to both antennas. Lastly, the design must handle an undesired notch frequency band resulting from the coupling between the antennas and the metasurface. In [25], a quad-band operation for 2G/3G/4G/5G is obtained by means of heavy optimization tasks of a multi-resonant asymmetric dipole, a T-shaped patch, nine parasitic elements, and the feeding line. Nevertheless, the radiation patterns suffer from serious instabilities.

Another way to achieve a multiband, directive operation is to employ a single radiating element covering the whole frequency range of interest associated with a multiband AMC. In [26,27,28], dual-band AMCs are associated with coplanar waveguide radiating elements. In these works, the AMC provides a reflection coefficient phase $\varphi_{r}=0{}^{\circ}$ at two different frequencies. The bow-tie shape has also been found to work well with AMCs [10]. In multiband designs, this radiating element is particularly interesting because it also allows the dual-polarization operation when two of them are placed orthogonally over the AMC, resulting in versatile antenna solutions [29,30].

It is shown in this article that a dual-band AMC can be associated with a wideband radiating element to provide a triple-band operation. This combination was previously used in [31] to design a triple-band, linearly polarized bow-tie antenna that exploits two resonances of an AMC for the Wi-Fi 2.4/5 and 5G standards. However, [31] does not study the principles behind the use of AMCs in this manner. A methodology enabling a number of operating frequency bands with considerable bandwidths is not proposed and the obtained 3 dB gain bandwidths are narrow, not covering the whole of the bands of interest. To the best of the authors’ knowledge, the present work is the first to propose a methodology in which the frequency and width of three operating frequency bands are controlled by adjusting the AMC and its spacing from the radiating element.

This study considers that the radio link between two devices should be optimal. Therefore, the figures of merit we focus on are the pattern stability within each frequency band of interest and the broadside realized gain which intrinsically computes the impedance mismatch. Regarding the frequency bands, European 4G/5G and Wi-Fi 2.4/5/6E are addressed, so that a wireless communication link may be established between a hotspot cell and the coverage tier in a single system. Thus, the system must cover three bands: 2.4–2.7 GHz, 3.4–3.8 GHz, and 5.17–6.45 GHz, respectively referred to as LB, MB, and UB hereafter. As already mentioned, obtaining stable, unidirectional, high-gain patterns in multiple bands with wide bandwidths is challenging. In this paper, we propose a methodology that, from a radiating element and an AMC both widely reported in the literature, leads to an antenna topology that outperforms prominent works in terms of gain, bandwidth, pattern stability, low profile, and simplicity.

## 2. Materials and Methods

### 2.1. Working Principle

Single-band AMCs emulate a PMC behavior at a specific frequency where a resonance occurs and the reflection phase $\varphi_{r}=0{}^{\circ}$. They also mimic a near-PEC behavior far from resonances where $\varphi_{r}\left(f\right)$ asymptotically goes to ±180°. Therefore, by placing a wideband radiating element at a quarter-wavelength distance from a single-band AMC in the highest band, which implies a distance much smaller than a wavelength for the lowest band, it is possible to achieve a stable, unidirectional, high-gain pattern in a dual-band operation with wide bandwidths.

The bands LB, MB, and UB should be addressed. A single-band AMC with a moderate bandwidth of 45.2% (2.40–3.80 GHz) would be sufficient to provide a near-PMC behavior in the bands LB and MB. However, since a near-PEC behavior must also be provided throughout the whole UB, the AMC reflection phase $\varphi_{r}\left(f\right)$ must approach ±180° in the interval between the MB highest frequency (3.80 GHz) and the UB lowest frequency (5.17 GHz). Therefore, instead of a single-band AMC with moderate bandwidth and soft phase response, a dual-band AMC with sharp phase response is required.

This dual-band AMC should emulate the PMC behavior at two specific frequencies close to each other and a near-PEC behavior far from resonances. When the AMC and the radiating element are spaced by a quarter-wavelength distance in the highest band, which implies a distance much smaller than a wavelength for the two lowest bands, it is possible to achieve a stable, unidirectional, high-gain pattern in a triple-band operation with wide bandwidths. Figure 2 depicts this principle considering the bands LB, MB, and UB.

In the present methodology, the only constraint related to the radiating element is that it must work well in the three bands of operation, presenting stable radiation patterns with maximum radiation oriented towards each broadside direction.

### 2.2. Choosing and Adjusting the Dual-Band AMC Unit Cell

Several dual-band shapes were compared in [32]. We chose the well-known double square over a ground plane for the width of each of its two bands, for the frequency ratio between them, and to emphasize the fact that the presented methodology can work with a simple shape. Figure 3a shows the unit cell geometry as well as its parameters.

To obtain further insight about the operation of the double-square unit cell, a parametric study was performed with the transient solver of CST Studio Suite, with a waveguide port and proper PEC and PMC boundary conditions to emulate an infinite array, as illustrated in Figure 3b. In this configuration, a transverse electromagnetic (TEM) wave illuminates the surface at normal incidence [33]. The black lines in Figure 4 show the reflection coefficient phase $\varphi_{r}\left(f\right)$ considering the parameters in Table 1.

In Figure 4a, $g_{1}$ is varied. The dependent parameter is the width of the inner square, thereby allowing the other parameters to be unchanged (when $g_{1}$ increases, the inner square decreases, keeping all other parameters unchanged). As $g_{1}$ increases, the frequency of the first resonance increases, with a minor increase in the frequency of the second resonance. Moreover, low values of $\varphi_{r}$ (for instance, −140°) are reached at slightly higher frequencies, indicating that the near-PEC behavior is slightly shifted upward. In Figure 4b, $g_{2}$ is increased while the width of the inner square decreases, again keeping the other parameters unchanged. As $g_{2}$ increases, the frequency of the second resonance also increases, as well as the operational bandwidth around it (the bandwidth here is defined by the condition $-90{}^{\circ}<\varphi_{r}<+90{}^{\circ}$). The operational bandwidth around the first resonance and its frequency slightly decrease. The near-PEC behavior again shifts upward in frequency. Increasing the width of the strip between gaps s while decreasing the width of the inner square works similarly to the $g_{2}$ case (not shown). In Figure 4c, the thickness $t_{h}$ is varied. As $t_{h}$ increases, both resonant frequencies decrease, the operational bandwidth around the first resonance increases, and that around the second resonance decreases. Lastly, when the periodicity p increases while the other parameters remain the same (only the width of the inner square increases), the entire graphs shrink to the left, decreasing both resonant frequencies and bandwidths (not shown). A similar behavior is seen for an increase in the relative permittivity $\varepsilon_{r}$.

From this parametric study, we deduce that the thickness $t_{h}$ is the parameter that acts the most on the operational bandwidth around the first resonance. Adjusting $t_{h}$ also impacts the operational bandwidth around the second resonance, but the latter may be compensated by means of $g_{2}$ or s. Furthermore, it is simple to control the frequency of the first resonance through $g_{1}$ with a minor impact on the rest of the phase response. However, it is not possible to control the frequency of the second resonance without affecting other features, since no parameter affects this frequency in an isolated manner. After these remarks, the following methodology is proposed to design the unit cell:

Based on [32], start with a unit cell with a periodicity $p=0.25\;\lambda_{l}$, gaps $g_{1}=0.02\;\lambda_{l}$ and $g_{2}=0.01\;\lambda_{l}$, and strip width $s=0.03\;\lambda_{l}$, where $\lambda_{l}$ is the wavelength at the frequency of the first resonance (as in [32], use a material with substrate thickness $t_{h}=3.15\;\mathrm{mm}$, relative permittivity $\varepsilon_{r}=2.2$, and dissipation factor tanδ=0.0009);Consider changing the material to one that has a more appropriate relative permittivity $\varepsilon_{r}$ if the second resonance is not close to the desired frequency;Consider changing the material to one that has another substrate thickness $t_{h}$ if the operational bandwidth achieved around the first resonance is not proper;Adjust the gap $g_{2}$ and the strip width s to achieve the desired bandwidth around the second resonance;Adjust the periodicity p to place the second resonance at the desired frequency;Adjust the gap $g_{1}$ to place the first resonance at the desired frequency; to achieve a fine adjustment in both resonance frequencies, some iterations between this and the previous step may be required.

Following this methodology, the unit cell defined in Table 1 was designed. The black line in Figure 5 shows the reflection phase $\varphi_{r}$ of this unit cell, which actually does not reach −180° in the UB. Figure 5 also shows the phase differences $\varphi_{t}$ between the reflected and direct electric fields obtained in a plane placed at different distances h from the unit cell. These curves are obtained by adjusting in CST Studio Suite the reference plane of the waveguide port represented in Figure 3b. In the final design, the condition $-120{}^{\circ}+2N\pi <\varphi_{t}<+120{}^{\circ}+2N\pi$ found in [4] is used to define the distance between the antenna and the reflector plane. We see in Figure 5 that for h=10 mm the areas where $-120{}^{\circ}<\varphi_{t}<+120{}^{\circ}$ (highlighted) encompass the bands LB, MB, and UB. This distance is around $\lambda_{h}/5$, where $\lambda_{h}$ is the wavelength at the UB center frequency (5.81 GHz). Changing h affects the path phase delay $\varphi_{p}\left(f\right)=-2kh$ and ultimately shifts in frequency the regions where the phase difference meets the condition $-120{}^{\circ}<\varphi_{t}<+120{}^{\circ}$.

### 2.3. Choosing and Adjusting the Radiating Element

Aiming at showing that the presented methodology can work with simple shapes, a bow-tie was used as the radiating element due to its simple planar shape and well-known behavior. Despite its wideband input-impedance behavior, the conventional, triangular bow-tie shape presents pattern instabilities that make energy deviate from the broadside direction at some frequencies [34]. Hence, aiming to enhance the radiation stability towards each broadside direction in the operating frequency bands (in agreement with the only constraint related to the radiating element in the present methodology; see Section 2.1), a rounded-edge bow-tie shape was adopted in which four radially aligned grooves are inserted. A detailed study on a grooved bow-tie without the presence of a reflector, including the radiation patterns and the currents over the antenna, is presented in [35]. Without the reflector, an optimal input impedance $Z_{ref}=185\;Ω$ is found. The introduction of a reflector, however, will change the input impedance, as expected. It is well-known in the literature that the bow-tie flare angle has a direct influence on the input impedance [36]. Hence, the antenna of [35] was adjusted, assuming the aspect of Figure 6.

The isolated antenna (without any reflector) of Figure 6 was simulated with the transient solver of CST Studio Suite using a discrete port with reference impedance $Z_{ref}=151\;Ω$. Table 2 shows the parameters of this antenna.

Figure 7 shows the simulated results for the standalone antenna. From 2.40 GHz, the reflection coefficient magnitude is $\left|\Gamma \right|\leq -8.4\;\mathrm{dB}$. A better $\left|\Gamma \right|$ could be achieved by increasing the length of the radiating parts, at the cost of hampering the broadside realized gain at high frequencies, or by using a matching circuit. However, since this study is focused on the broadside realized gain and the pattern stability within each frequency band of interest, this reflection coefficient magnitude $\left|\Gamma \right|$ is considered satisfactory. The broadside realized gain is above 2 dBi from 2.40 to 6.45 GHz, except for a drop around 4.40 GHz, which is mainly due to the insertion of the grooves. The antenna is suitable for the final structure because this drop is outside the bands of operation.

### 2.4. Putting the AMC and the Radiating Element Together

We simulated the bow-tie defined in Table 2 spaced by h=10 mm from AMCs having different numbers of unit cells as defined in Table 1 with CST Studio Suite using a discrete port with reference impedance $Z_{ref}=151\;Ω$. Figure 8 shows the results for three cases: the bow-tie over a 6×6, 8×8 and 10×10-cells AMC.

Mainly in the MB and UB bands, the reflection coefficient magnitude presents significant changes as the number of cells varies, indicating that both bands are affected by the finitude of the AMC. One reason reported in the literature for this behavior is the reflection of surface waves in the edges of the AMC [37]. In the broadside realized gain, illustrated in Figure 8b, the instabilities in the MB band change as the number of cells varies. In addition, a drop in gain in the UB clearly shifts across the frequency domain, showing that it depends on the finitude of the AMC. For the three configurations, this gain drop causes a significant performance deterioration and should be addressed. Figure 9 shows the magnitude of the ${â}_{y}$-component of the currents on the structure with the 8 × 8-cell AMC at 3.44 and 5.54 GHz.

It is visible that some patches resonate out of phase with adjacent patches. Specifically, at 3.34 GHz, entire columns of patches are out of phase with their adjacent columns. At 5.54 GHz, the patches of some columns resonate in couples aligned in the y-dimension. Therefore, breaking the even parity of the number of cells in this dimension should change this configuration. In Figure 10, the simulated results for a 7 × 8-cell AMC are shown. The results for the 8 × 8-cell AMC are repeated for reference.

In the reflection coefficient magnitude of Figure 10a, no useful information about the band UB is obtained but it is visible that the drop in the reflection coefficient magnitude at 3.44 GHz shifted downward out of the MB band. In the broadside realized gain of Figure 10b, the drop in gain at 5.54 GHz, directly addressed in this attempt, is considerably softened from 6.7 to 9.8 dBi, from the version with the 8 × 8-cell AMC to that with the 7 × 8-cell AMC. Additionally, the drop in gain at 3.44 GHz, similar to the reflection coefficient magnitude, shifted downward out of the MB band. As such, all the LB, MB, and UB bands are covered with a good gain level.

Before inserting a balun and foam bricks for mechanical support so as to fabricate the device, we were interested in observing how the performance changed with the spacing h between the AMC and the bow-tie. Figure 11 shows the reflection coefficient magnitude and the broadside realized gain when the spacing h varies from 5 to 20 mm. For reference, the curve styles used here match those of Figure 5 for respective values of h. We can see in Figure 11a that the reflection coefficient magnitude is highly sensitive to changes in the spacing h in all the addressed bands. This behavior is expected since the interferences of the fields around the bow-tie input terminals change according to h, presenting an impact on the input impedance of the device. Figure 11b shows that, as the separation h between the bow-tie and the AMC increases, the frequency ranges in which the broadside realized gain is maximum are shifted downward. This behavior is consistent with that seen in Figure 5, where the regions in which the phase difference $\varphi_{t}$ respects the condition $-120{}^{\circ}<\varphi_{t}<+120{}^{\circ}$ shift downward in frequency when h increases. Specifically, the value of the spacing between the AMC and the bow-tie that covers all the LB, MB, and UB bands with a good broadside gain is h=10 mm, which coincides with the value in Figure 5 that respects the condition $-120{}^{\circ}<\varphi_{t}<+120{}^{\circ}$ for all the bands.

Next, the bow-tie defined in Table 2 was etched on an Arlon DiClad 880 substrate layer whose thickness is 0.76 mm, with relative permittivity $\varepsilon_{r}=2.20$ and tanδ=0.0009. A 4.9 mm thick AMC composed of 7 × 8 cells defined in Table 1 was modeled with two sheets of Arlon DiClad 870 material, each one measuring 160 × 142 × 2.45 mm^3^, $\varepsilon_{r}=2.34$, and tanδ=0.0013. Figure 12 shows the final CST design.

A 3:1 impedance ratio exponential taper balun [38] etched on the same material as that of the bow-tie allows the feeding of the structure with a 50 Ω SMA connector. The inset of Figure 12 details the hole used to pass the balun through the AMC. A foam brick ($\varepsilon_{r}=1.1$, tanδ=0.005) surrounds the balun at the back of the device, providing mechanical support. Moreover, the distance h=10 mm between the radiators and the AMC is ensured by two smaller foam bricks, positioned as shown in Figure 12 to avoid hampering the results.

The CST transient solver was used to simulate the device fed by a waveguide port placed at the input of the SMA connector. The reflection coefficient magnitude and the broadside realized gain are also shown in Figure 13. For comparison purposes, a simulation in which the dual-band AMC is substituted by a PEC reflector of same total area, placed at the same distance h=10 mm over the antenna, was also performed. In addition, we repeat in Figure 13 the results for the standalone bow-tie. We also took advantage of this moment to simulate the radiation efficiency of the studied devices.

For the dual-band AMC, peak values of 8.6, 9.8, and 10.7 dBi, were respectively achieved in terms of broadside realized gain for the LB, MB, and UB. The reflection coefficient magnitudes were better than −8.4, −11.2, and −6.2 dB in the LB, MB, and UB. Again, a better reflection coefficient magnitude $\left|\Gamma \right|$ is possible with a matching circuit if a specific application requires it. In this study, we focused on the pattern stability and the broadside realized gain, which intrinsically computes the impedance mismatch. For the PEC case, peak values of 6.2, 9.7, and 8.9 dBi of broadside realized gain are respectively provided for the LB, MB, and UB, which means that the proposed structure provides an enhancement of 2.4, 0.1, and 1.8 dB, respectively. Moreover, magnitudes better than −2.8, −9.0, and −7.1 dB are verified for the reflection coefficient in the LB, MB, and UB, respectively. In terms of radiation efficiency, we see that the standalone bow-tie presents values close to 1 in all the LB, MB, and UB bands. The introduction of an AMC allows the efficiency to remain as high as 0.98, 0.95, and 0.95 in the LB, MB, and UB bands, respectively. These results show that the introduction of the AMC does not have a significant impact in this matter, even though it presents a lossy dielectric sheet.

In what follows, we show that, for a fixed spacing h between the AMC and the radiating element, the performance of the whole structure is mainly controlled through changes in the AMC design, as long as the radiating element works well in the operating frequency bands. In Figure 14, a variation in g1 is shown. As g1 increases, the peak of broadside realized gain around the LB significantly shifts upward. The same happens for the peaks around the MB and the UB, but in a soft manner. In the MB, the broadside realized gain is limited by the performance of the antenna, whose gain presents a drop around 4.40 GHz. The behavior seen in Figure 14 agrees with that seen in Section 2.2 when the gap g1 is varied, confirming that the AMC controls the performance of the whole device. Figure 14 also shows the phase difference $\varphi_{t}$ in the plane located 10 mm over the unit cell, which relates to the distance between the AMC and the bow-tie in the final device. The gain peaks do not occur at the same frequencies at which $\varphi_{t}=0$. This happens because $\varphi_{t}$ is calculated in conditions of normal incidence of TEM waves and infinite dimensions of the AMC. In the case of a finite-array AMC over which an antenna is located:

The AMC is in the antenna near-field region and is not illuminated by a plane wave;Some coupling effects between antenna and AMC may take place;The finitude of the AMC creates extra surface wave resonances [37].

Consequently, when the complete device is simulated, the broadside gain maxima do not occur exactly at the same frequencies as those of the zeros of the phase difference $\varphi_{t}$ for the unit cell. Nevertheless, when the unit cell parameters vary, the relative shifts of these maxima and zeros are similar, as seen in Figure 14. Hence, if the operational frequency bands are not centered around the desired frequencies in the first simulation of the complete device, adjustments may be made in the unit cell in order to correct them.

We also studied variations in the other AMC parameters, which showed that they also control the response of the complete structure, confirming the fact that the AMC rules the performance of the structure in terms of radiation in the broadside direction.

## 3. Results

The prototype (Figure 15) was made of the different layers whose materials are specified in Section 2.4 and that were etched with an LPKF ProtoLaser S4 machine.

The measured reflection coefficient magnitude is shown in Figure 16a. A particularly good agreement occurs in the MB, where $\left|\Gamma \right|\leq -10.0\;\mathrm{dB}$. The measurements also agree well in the LB and UB, presenting slightly better values than in simulations (≤−11.5 dB for the LB and ≤−7.4 dB for the UB).

Figure 16b shows the measurement of broadside realized gain, in which a particularly good agreement with simulations is seen again. The following ranges of values were achieved in the measurements for the LB, MB, and UB bands, respectively: 7.4–8.0, 7.6–9.1, and 8.8–10.5 dBi. In the simulations, the ranges were 8.0–8.6, 8.1–9.8, and 8.1–10.7 dBi. The measured broadside realized gain shows 3 dB bandwidths of 34.4% for the lower operating band (from 2.00 to 2.83 GHz), 19.7% for the middle band (from 3.20 to 3.90 GHz), and 31.0% for the upper band (from 4.90 to 6.70 GHz), which means that the antenna, in addition to being multiband, is also wideband.

Figure 17 shows the radiation pattern in the three frequency bands of operation. The co-polarized component agrees well with simulations in both E- and H-planes. The half-power beamwidth of both planes is stable within each of the bands. Values of 64.0°, 59.5°, 53.5°, 56.5°, 27°, and 26° are seen from 2.4 to 6.45 GHz for the E-plane, versus 70.0°, 85.0°, 76.5°, 74.0°, 50.5°, and 45.5° for the H-plane. For all frequencies, the front-to-back ratio is better than 16 dB in the simulations and 19 dB in the measurements.

The cross-polarized level is below −18 dB for the H-plane in both measurements and simulations. In the E-plane, the cross-polarized level is below −25 dB in the measurements, while in simulations it tends to −∞ dB. The measured results and the agreement with simulations validate the presented methodology for providing a unidirectional, stable, high-gain pattern in the three desired frequency bands of operation. As such, the technique is able to cover the addressed standards (4G/5G and Wi-Fi 2.4/5/6E in Europe) and to enable wireless data exchange between hotspot cells and the coverage tier in a single system.

## 4. Discussion

The methodology presented in this article, concerning the design of low-profile triple-band antennas with stable, unidirectional, high-gain radiation patterns using dual-band AMCs, exploits the two main modes of operation that may be found in AMCs: one similar to a PMC and the other to a PEC reflector. In this paper, we detailed how to adjust a simple wideband radiating element and a simple dual-band AMC, as well as the spacing between them, in order to obtain a triple-wideband operation. Moreover, we performed analysis of the currents over the AMC in order to choose the best number of cells. This methodology was validated with the realization of a prototype for the European standards of 4G/5G and Wi-Fi 2.4/5/6E.

The prototype presents a thickness of $0.12\;\lambda_{l}$, where $\lambda_{l}$ is the wavelength in the lowest operational frequency, i.e., 2.40 GHz. The balun used in this work, oriented orthogonally to the antenna plane, is not considered in this calculation, since baluns in the same plane of the antenna can be used (the orthogonally oriented balun was used for simplicity). Simulations of the radiation efficiency have shown that the introduction of the AMC does not degrade such a figure of merit in a considerable manner, even though it presents a lossy dielectric sheet. Radiation efficiencies better than 0.98, 0.95, and 0.95 were found in the LB, MB, and UB bands, respectively. The measured broadside realized gain shows 3 dB bandwidths of 34.4% for the lower operating band (from 2.00  to 2.83 GHz), 19.7 % for the middle band (from 3.20  to 3.90 GHz), and 31.0% for the upper band (from 4.90  to 6.70 GHz), which means that the antenna, in addition to being multiband, is also wideband. Moreover, it achieves peak gain values of 8.0, 9.1, and 10.5 dBi in the frequency bands of 2.40–2.70, 3.40–3.80, and 5.17–6.45GHz, respectively. Table 3 summarizes the features of prominent multiband antenna works.

We can see that the only works comparable to ours in terms of bandwidth are [25,28], and maybe [23]. Moreover, only [25,26,28] present thickness lower than ours. However, the device of [25] is complex and its radiation patterns suffer from serious instabilities (the maximum of gain points at different directions at each frequency). Furthermore, in [23,26,28], the peak gain is about 5 dB below ours. As already mentioned, [31] exploits a bow-tie and a dual-band AMC in a triple-band operation, similar to our work. Besides a thickness larger than ours, this work also presents narrow widths for each of the bands and peak gains below ours (from 2.6 to 0.9 dB below). We can also see in Table 3 that the in-band gain variations of our 
device are softer than the others. For instance, [23] 
presents ranges as large as 2.1 dB, [25] as large as 5.8 dB, [26] as large as 3.5 dB, and [31] as large as 3.6 dB, while our work presents a maximum range of 1.7 dB (in the UB). All of that being said, the design methodology proposed in the present article enables optimized performances to be obtained. To the best of the authors’ knowledge, no previous work exists that simultaneously rivals ours in all the features combined (bandwidths, gain, pattern stability, low profile, and simplicity of the employed structures).

Finally, it was demonstrated that, if the radiating element works well in the bands of interest, the AMC controls the performance of the final device. As seen in [29,30], the use of a couple of orthogonal bow-ties associated with a dual-band AMC can lead to dual-band, dual-polarized antennas. In addition, a couple of orthogonal bow-ties also worked well in a dual-band, circularly polarized device that exploits both the near-PEC and the near-PMC behaviors of a single-band AMC [39]. Therefore, we believe that the presented methodology may lead to a triple-band, dual-polarized device if a dual-band AMC is employed together with a couple of orthogonal bow-ties, which is stated here as a future work.

## Figures and Tables

**Figure 1 sensors-23-01920-f001:**
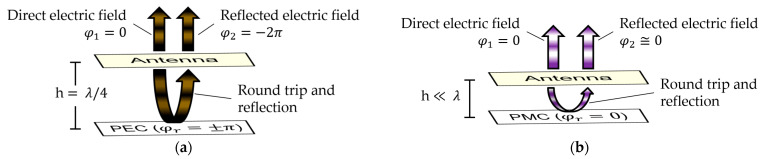
Optimization of gain in the broadside direction: (**a**) antenna distanced by h=λ/4 from a PEC reflector; (**b**) antenna distanced by h≪λ from a PMC reflector.

**Figure 2 sensors-23-01920-f002:**
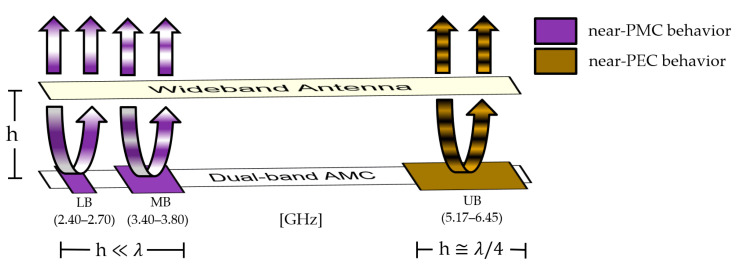
Working principle: the dual-band AMC works as a near-PMC reflector around two resonances and as a near-PEC reflector far from resonances.

**Figure 3 sensors-23-01920-f003:**
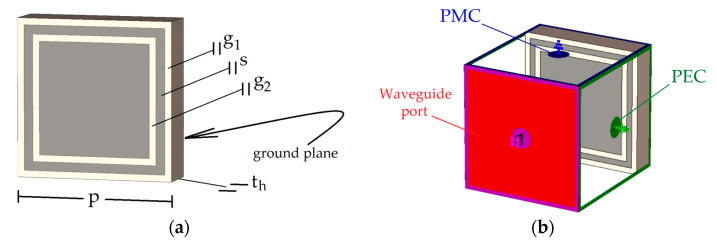
AMC unit cell: (**a**) parameters; (**b**) port and boundary conditions.

**Figure 4 sensors-23-01920-f004:**
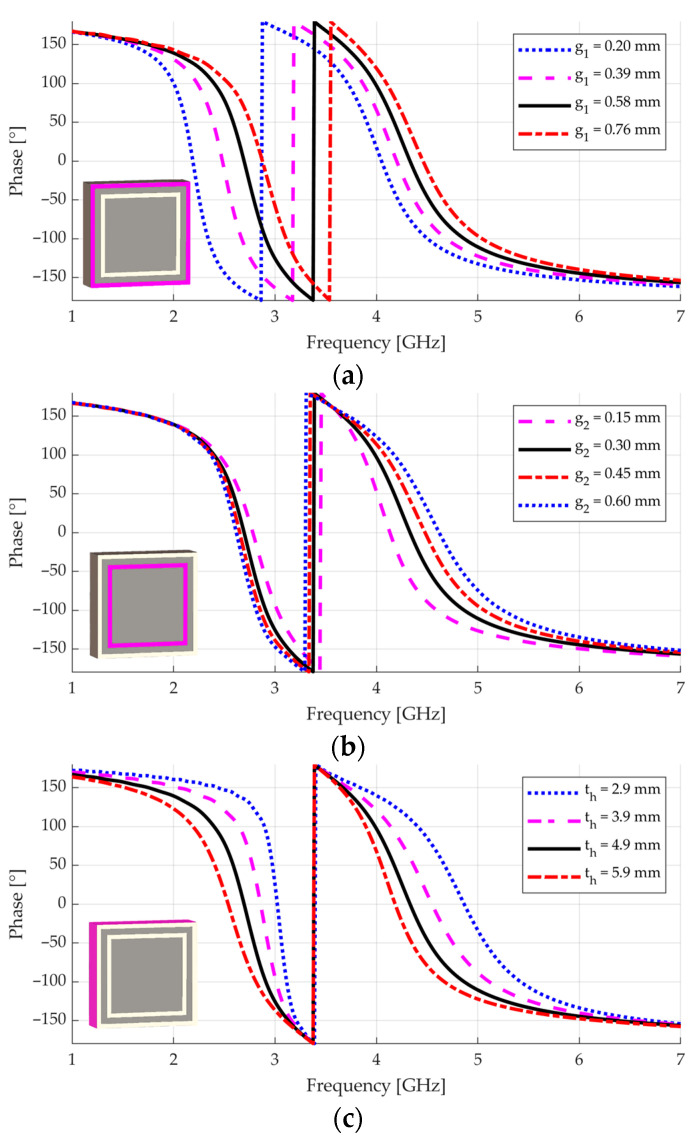
Reflection phase of the double-square element when a single parameter is varied: (**a**) $g_{1}$; (**b**) $g_{2}$; (**c**) $t_{h}$. For better visualization, the insets show the part of the structure related to the parametric variation.

**Figure 5 sensors-23-01920-f005:**
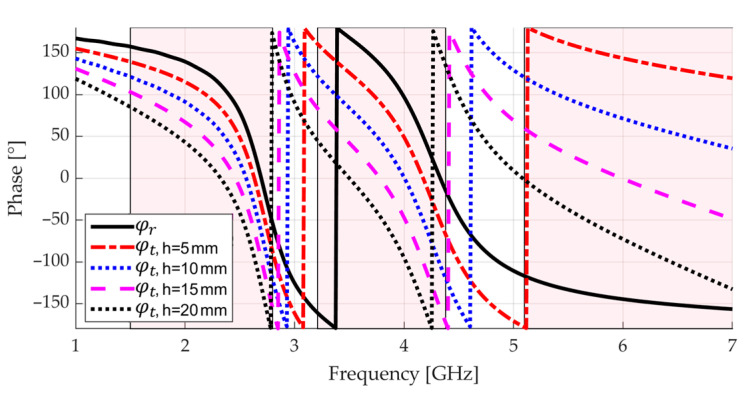
Reflection phase of the final unit cell in its own plane ($\varphi_{r}$) and in the plane of an antenna spaced by a distance h ($\varphi_{t}$). The highlighted areas show when $\left|\varphi_{t}\right|<120{}^{\circ}$ for h=10 mm.

**Figure 6 sensors-23-01920-f006:**
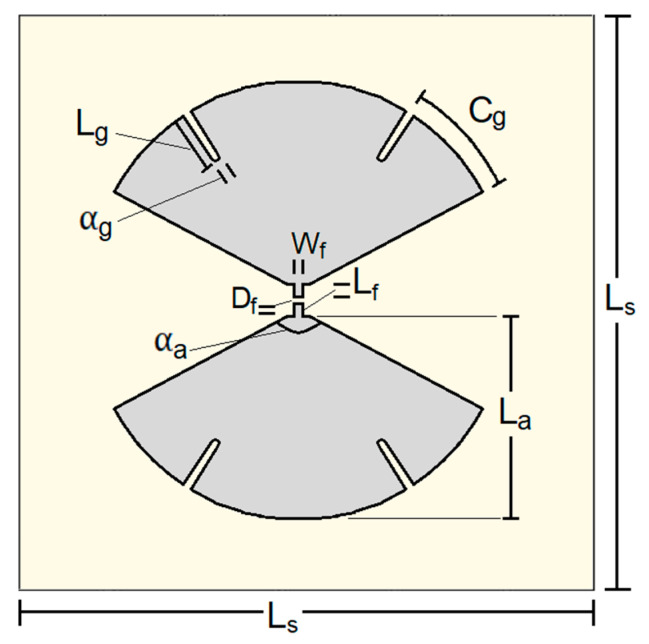
Rounded-edge bow-tie with grooves and its parameters.

**Figure 7 sensors-23-01920-f007:**
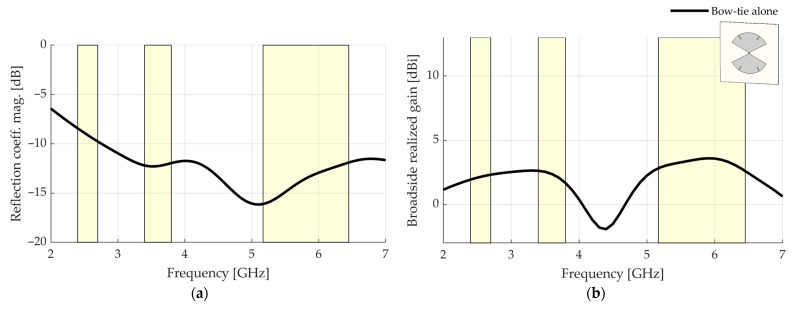
Simulated results for the standalone antenna (without any reflector): (**a**) reflection coefficient magnitude; (**b**) broadside realized gain. The bands of interest are highlighted.

**Figure 8 sensors-23-01920-f008:**
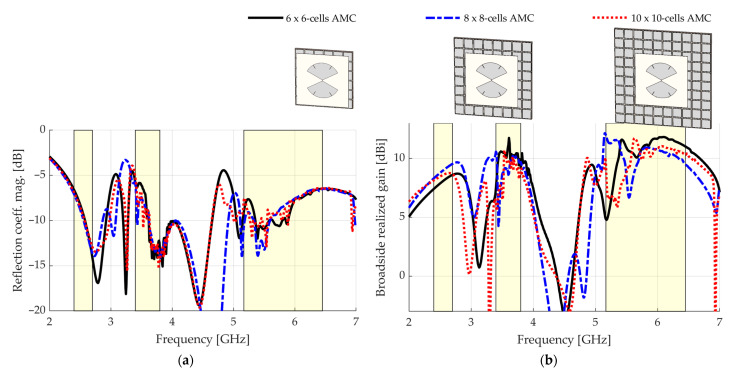
Simulated results when the number of cells varies: (**a**) reflection coefficient magnitude; (**b**) broadside realized gain.

**Figure 9 sensors-23-01920-f009:**
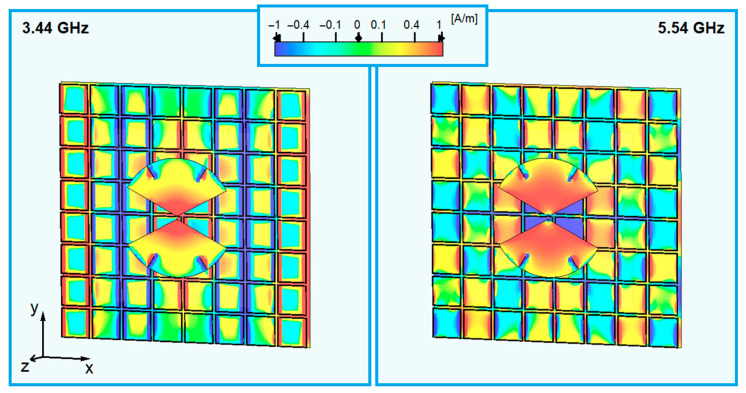
Magnitude of the ${â}_{y}$-component currents at 3.44 and 5.54 GHz on the 8 × 8-cell AMC.

**Figure 10 sensors-23-01920-f010:**
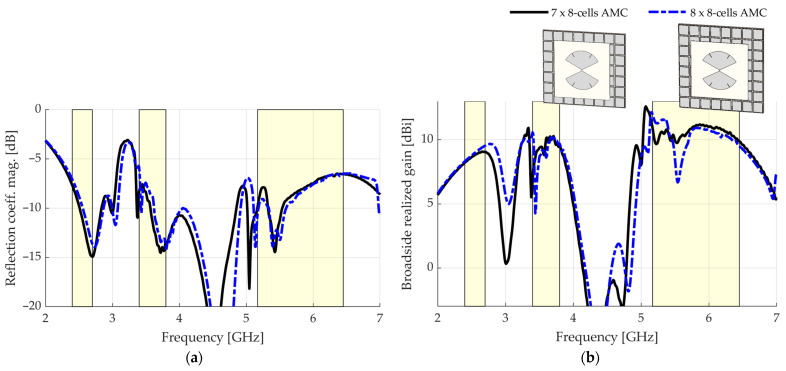
Simulated results for a 7 × 8-cell AMC: (**a**) reflection coefficient magnitude; (**b**) broadside realized gain.

**Figure 11 sensors-23-01920-f011:**
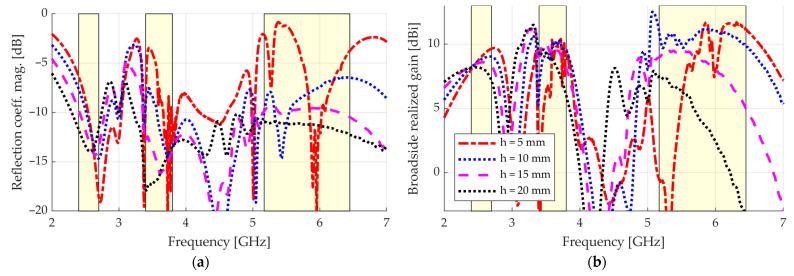
Simulated results when the spacing h is varied.: (**a**) reflection coefficient magnitude; (**b**) broadside realized gain.

**Figure 12 sensors-23-01920-f012:**
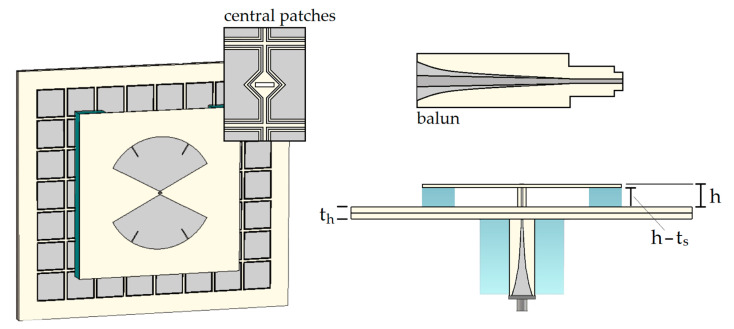
Final design: the AMC unit cell parameters are defined in Table 1; the bow-tie parameters are shown in Table 2; the inset shows the hole to pass the balun in the central patches.

**Figure 13 sensors-23-01920-f013:**
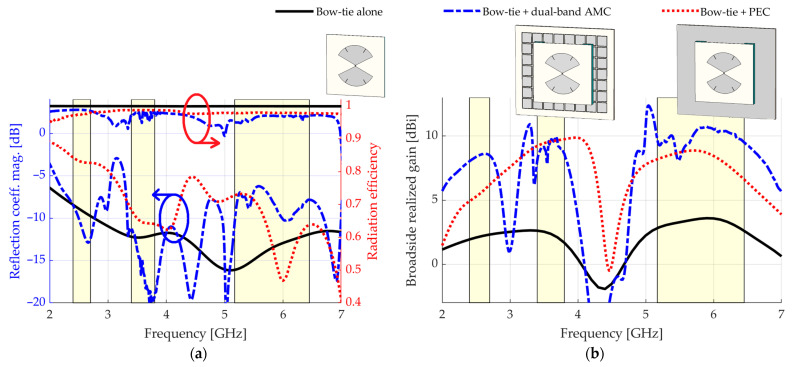
Simulated results for the standalone antenna, the antenna backed by the dual-band AMC, and the antenna backed by a PEC reflector: (**a**) reflection coefficient magnitude and radiation efficiency; (**b**) broadside realized gain.

**Figure 14 sensors-23-01920-f014:**
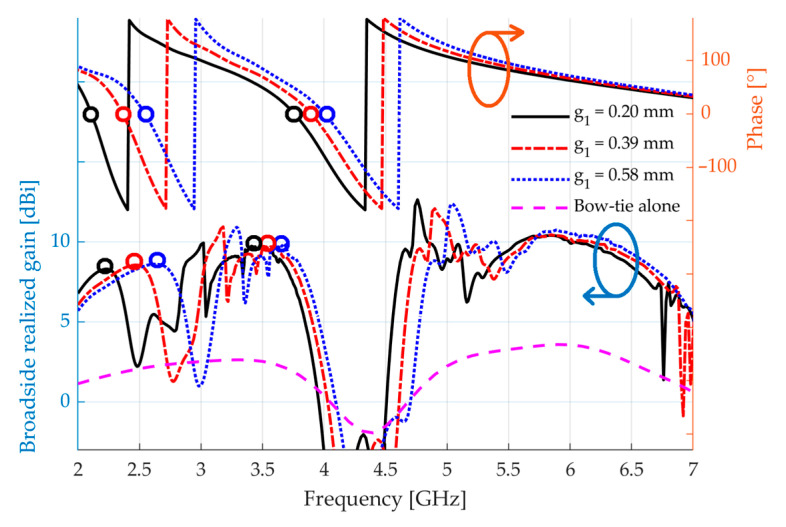
Broadside realized gain of the final structure when g1 is varied. The phase difference $\varphi_{t}$ between the direct and reflected electric fields over the unit cell, considering h=10 mm, is also shown. Dots indicate each $\varphi_{t}=0{}^{\circ}$ and their respective gain peaks. Their frequencies do not coincide, but their relative shifts, when g1 varies, do.

**Figure 15 sensors-23-01920-f015:**
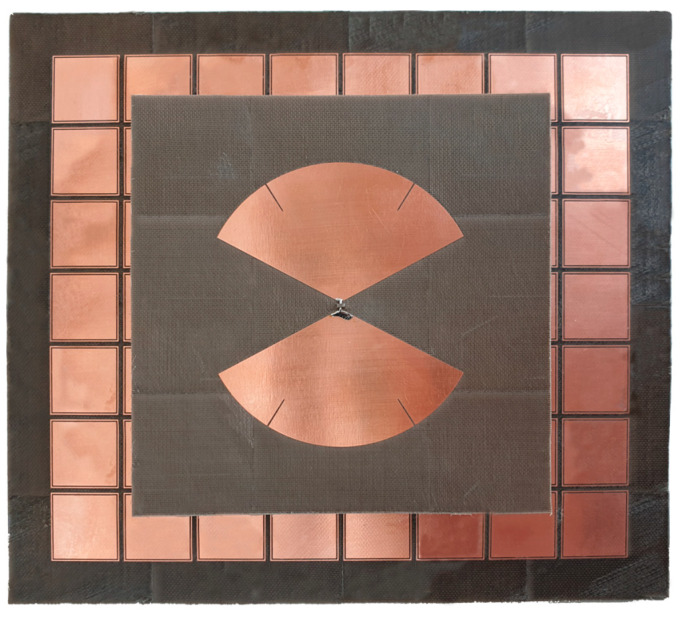
Realized antenna with AMC and balun.

**Figure 16 sensors-23-01920-f016:**
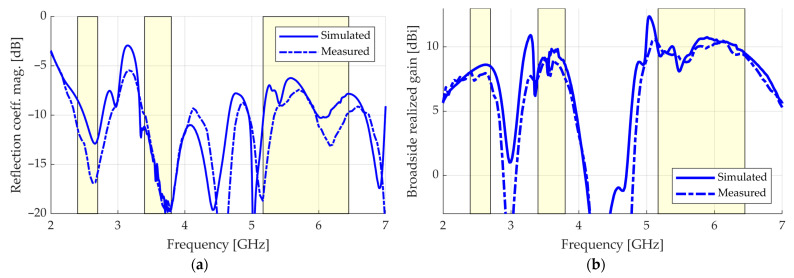
Results: (**a**) reflection coefficient magnitude; (**b**) broadside realized gain.

**Figure 17 sensors-23-01920-f017:**
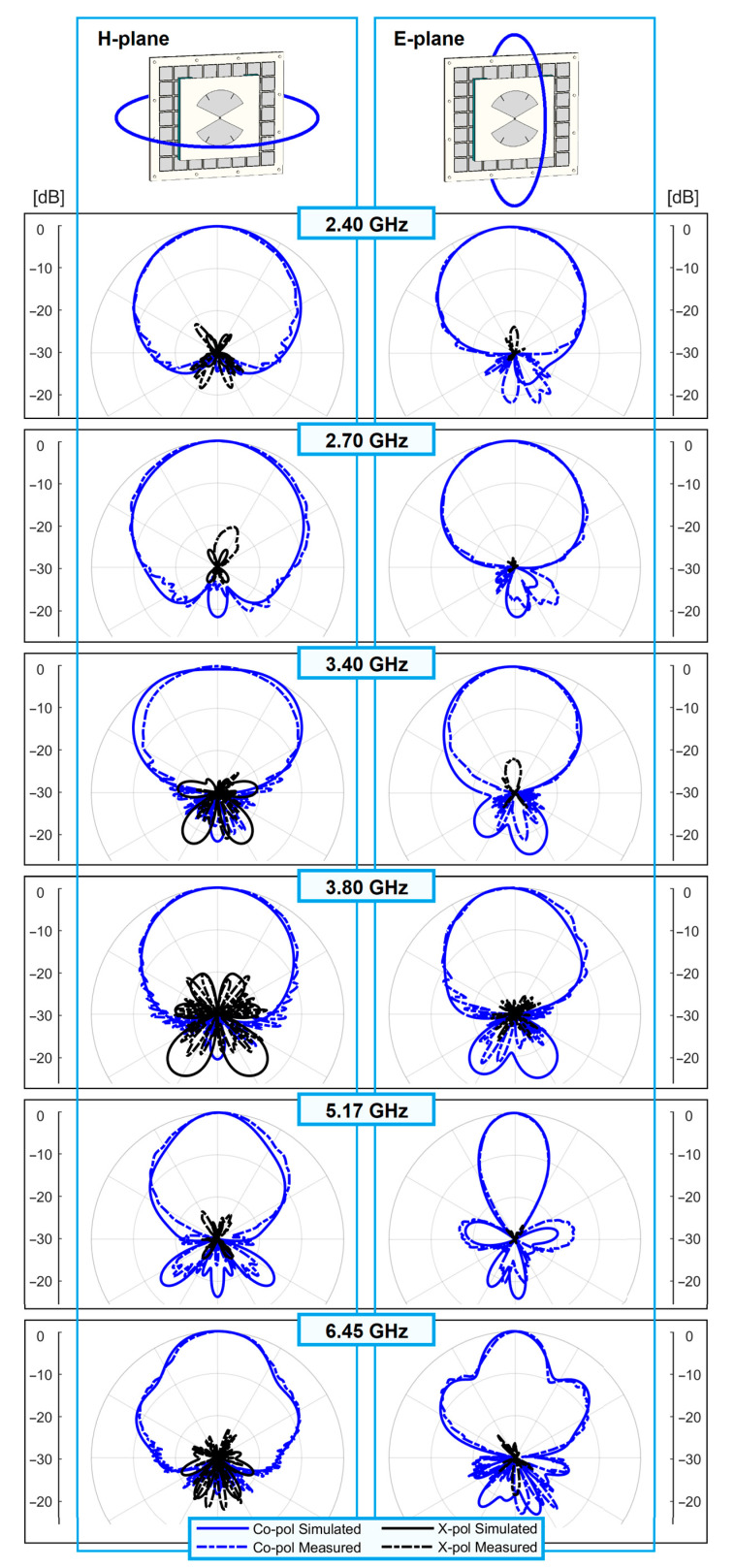
Radiation patterns at the limit frequencies of the LB, MB, and UB, and H- and E-planes.

**Table 1 sensors-23-01920-t001:** Parameters of a double-square unit cell.

Symbol	Description	Value
$\varepsilon_{r}$	substrate relative permittivity	2.34
tanδ	substrate loss tangent	0.0013
$g_{1}$	outer gap	0.58 mm
$g_{2}$	inner gap	0.30 mm
s	strip between gaps	0.25 mm
p	periodicity	17.5 mm
$t_{h}$	substrate thickness	4.9 mm

**Table 2 sensors-23-01920-t002:** Antenna parameters.

Symbol	Description	Value
$\varepsilon_{r}$	substrate relative permittivity	2.2
tanδ	substrate loss tangent	0.0009
$L_{s}$	substrate length	94.5 mm
$t_{s}$	substrate thickness	0.76 mm
$L_{a}$	bow length	31.5 mm
$\alpha_{a}$	bow flare angle	124.4°
$L_{f}$	feeding strip length	0.80 mm
$W_{f}$	feeding strip width	0.60 mm
$D_{f}$	gap between feeding strips	0.76 mm
$L_{g}$	groove length	7.0 mm
$\alpha_{g}$	groove angular width	0.69°
$C_{g}$	groove angular position	30.0°

**Table 3 sensors-23-01920-t003:** Comparison between prominent multiband antenna works.

Work	Frequency Ranges [GHz]	3-dB Gain Bandwidth	Number of Sources	Where the Complexity Is	$\mathrm{Profile}\;[\lambda_{l}]$	Realized Gain [dBi]
[23]	1.86–1.92	3.2%	4	source	0.17	3.4–5.2
	2.30–2.65	14.1%				1.9–4.0
	3.40–3.80	11.1%				3.1–4.7
	5.30–6.92	26.5%				3.5–5.5
[25]	0.80–0.96	18.2%	2	source, feeding and parasitic elements	0.10	4.0–6.0
	1.70–2.70	45.4%				4.0–9.8
	3.30–3.80	14.1%				4.8–9.0
	4.80–5.00	4.1%				6.0–8.7
[26]	1.571	−	1	reflector	0.04	1.1 (peak)
	1.71–2.18	24.2%				4.1–5.9
	2.40–2.48	3.3%				3.9 (peak)
	5.17–5.84	12.2%				4.4–7.9
[28]	1.95–2.68	31.5%	1	reflector	0.07	6.8 (peak)
	3.46–3.93	12.7%				6.5 (peak)
	4.18–6.59	49.8%				7.3 (peak)
[31]	2.39–2.63	9.6%	1	reflector	0.17	2.7–5.6
	3.61–3.72	3.0%				2.9–6.5
	5.61–5.84	3.7%				7.0–9.6
This work	2.00–2.83	34.4%	1	reflector	0.12	7.4–8.0
	3.20–3.90	19.7%				7.6–9.1
	4.90–6.70	31.0%				8.8–10.5

## Data Availability

Not applicable.

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
