# Peer review of "A Low-Profile, Triple-Band, and Wideband Antenna Using Dual-Band AMC"

_sensors, 2023, doi:10.3390/s23041920_

Round 1

Reviewer 1 Report

1. Why authors have chosen a bow tie antenna? A simple monopole antenna with wide impedance bandwidth can do the work.

2 Please show radiation patterns of the antenna without AMC ( antenna alone)  in polar form at all three frequencies.

3. How you simulated the antenna alone structure for Fig. 7 ? ( asking about ground plane of antenna)

4 Please explain why you have chosen a discrete port (instead of waveguide port) for simulation.

5. Please explain more about phase difference ?t. Also, elaborate more about the +-120 degree criteria. How you simulated  ?t in CST?

Reviewer 2 Report

(1) Based on Fig. 11 (a), in the upper band of interest (from 5.17 ??? to 6.45 ???), the simulated reflection coefficient magnitude is not less than 10 dB (|?|>=−10.0 dB), which could not satisfy the general requirements for antenna design. Meanwhile, the reason why there is a large discrepancy between the simulated and measured reflection coefficient magnitudes in the upper band of interest needs to be explained in detail.

(2) Based on Fig. 11 (b), in the lower, middle and upper bands of interest, i.e., 2.4–2.7 GHz, 3.4–3.8 GHz and 5.17–6.45 GHz, the ranges of the simulated and measured broadside realized gain should be listed, instead of the simulated and measured peak gain values. For instance, in the upper bands of interest, i.e., 5.17–6.45 GHz, the range of the simulated broadside realized gain is between 8dBi and 10.7dBi.

(3) In Table 3, the comparison between the performances of the multiband antennas proposed by the authors and presented in the references should be NOT based on the peak gain values but based on the ranges of the broadside realized gain. The former does not make sense.

Reviewer 3 Report

This is an interesting work, only contains several minor concerns.

1. According to the measured results in Fig. 11, it seems that the proposed triple-band antenna does not show sufficient enough performance in the second and third bands. For instance, as shown in Fig. 11, the reflection in third band is -6 ~ -13dB, which swings quite large. Then, for the realized gain, the second band has a Gain of 8-10dBi, while the third band has a Gain of 8-11dBi. Therefore, the in-band variations of the realized gains in the second and third bands are really too large. 

2. Table 3 summarizes a comparison between the proposed antenna and some reported triple- and quaduple-band antennas. However, according to this table, what is the main advantages and innovation of the proposed antenna? It seems that no nortable advantages can be captured in current version of Table 3.

Reviewer 4 Report

The authors presented a Low-Profile, Triple-Band, and Wideband Antenna Using Dual-band AMC. The concept is exciting, and the simulation results are reasonably good, showing potentially strong reconfigurability. I have the following suggestions before accepting it for publication:

The idea of this paper is simple: Antenna+AMC reflector= High antenna gain. The AMC unit cell is too basic, which is a square loop structure!!!!

Introduction:

- The Introduction needs to be improved. The authors must explain some Metamaterial types. Frequency selective surfaces (FSSs) are considered one of them and studies their filtering behaviour when they act as stop-band transmission coefficient [1,2]. The gain enhancement techniques using FSS must also be clearly discussed when FSS acts as a single or multi-FSS reflector [3]. Please see these references, which may add value to the Introduction:

[1] A Wideband High-Gain Microstrip Array Antenna Integrated with Frequency-Selective Surface for Sub-6 GHz 5G Applications. Micromachines 202213, 1215. https://doi.org/10.3390/mi13081215.

[2] Gain Enhancement of a Dual-Band Antenna with the FSS. Electronics 202211, 2882. https://doi.org/10.3390/electronics11182882.

[3] Enhancing Gain for UWB Antennas Using FSS: A Systematic Review. Mathematics 20219, 3301. https://doi.org/10.3390/math9243301.

-I love the idea of drawing Figure 1. But it is not clear; Please make it precise.

2.2. Choosing and adjusting the dual-band AMC unit cell 

-I want to see the boundary condition of the proposed AMC unit cell.

3. Results 

-As I said, the AMC unit cell is too basic, a square loop. The authors should support the contributions of this paper by studying the parametric study of the proposed AMC square loop. In other words, the gap between the triple-band antenna and the AMC reflector should be denoted as an "Air-gap." However, this parameter plays a crucial role in antenna engineering gain enhancement. Please create a parametric study based on the "Air-gap" parameter and show its effect on the S-parameter and Gain, respectively.

The Gap parameter "Air-gap" values should be as follows:

1- Air-gap= 5mm.

2-Air-gap=10mm.

3-Air-gap=15mm.

4-Air-gap=20mm.

5-Air-gap=25mm.

To make it straightforward, please add two graphs as a parametric study of the "Air-gap" parameter, as mentioned above.

 -Moreover, as presented in the article, the number of AMC unit cells to structure the reflector is 8*8. Another parametric study should be conducted and named the "Amount of AMC cells." parameter and show its effect on the S-parameter and Gain, respectively.

The "Amount of AMC cells" parameter values should be as follows:

1- 6*6.

2-7*7.

3-8*8.

4-9*9.

5-10*10

To make it again clear, please add two graphs as a parametric study of the "Amount of AMC cells" parameter, as mentioned earlier.

-I want to see the simulated and measured radiation efficiency as follows:

a- Antenna with AMC-backed reflector.

b- Antenna without AMC-backed reflector.

-Please illustrate the current antenna distribution with and without the AMC reflector to see the current saturated on AMC unit cells (with/without).

4. Discussion

The Conclusions should be rewritten with the updated results above.

-That's all for me at this moment. However, the authors are required to revise the comments above carefully!

Best regards

Round 2

Reviewer 1 Report

All my queries are satisfactorily answered by the authors.

Author Response

The authors would like to thank the Reviewer 1 for the invaluable suggestions that  contribute to a better published article.

Reviewer 2 Report

My comments and concerns have been addressed properly. I consider that the revised manuscript could be accepted for publication.

Author Response

The authors would like to thank the Reviewer 2 for the invaluable suggestions that  contribute to a better published article.

Reviewer 4 Report

The authors have revised the given comments successfully, and I believe the article is ready now to be published in a reputational journal like Sensors. However, there are still typos and spacing errors that need to be carefully checked.

Best regards

Author Response

First of all, the authors would like to thank the Reviewer 4 for the invaluable suggestions that contribute to a better published article.

We have carefully checked the typos. English has also been extensively revised. The modifications are highlighted in the new version of the manuscript.